# Characterization of the Tumor Microenvironment and the Biological Processes with a Role in Prostatic Tumorigenesis

**DOI:** 10.3390/biomedicines10071672

**Published:** 2022-07-12

**Authors:** Cristina-Anita Ionescu, Mariana Aschie, Elena Matei, Georgeta Camelia Cozaru, Mariana Deacu, Anca Florentina Mitroi, Gabriela Isabela Baltatescu, Antonela-Anca Nicolau, Laura Mazilu, Liliana Ana Tuta, Ionut Ciprian Iorga, Alina Stanigut, Manuela Enciu

**Affiliations:** 1Chemical Carcinogenesis and Molecular Biology Laboratory, Institute of Oncology “Prof. Dr. Alexandru Trestioreanu”, 022328 Bucharest, Romania; anitaionescu7@gmail.com; 2Medicine Faculty, “Ovidius” University of Constanta, 1 Universitatii Street, 900470 Constanta, Romania; aschiemariana@yahoo.com (M.A.); deacu_mariana@yahoo.com (M.D.); lauragrigorov@gmail.com (L.M.); tutaliliana@yahoo.com (L.A.T.); ionut_iorga@yahoo.com (I.C.I.); asburlan29@yahoo.com (A.S.); iftimemanuela@yahoo.com (M.E.); 3Center for Research and Development of the Morphological and Genetic Studies of Malignant Pathology, “Ovidius” University of Constanta, 145 Tomis Blvd., 900591 Constanta, Romania; drcozaru@yahoo.com (G.C.C.); ank_mitroi@yahoo.com (A.F.M.); gabrielabaltatescu@yahoo.com (G.I.B.); ancanicolau@rocketmail.com (A.-A.N.); 4Clinical Service of Pathology, “Sf. Apostol Andrei” Emergency County Hospital, 145 Tomis Blvd., 900591 Constanta, Romania; 5Oncology Department, “Sf. Apostol Andrei” Emergency County Hospital, 145 Tomis Blvd., 900591 Constanta, Romania; 6Nephrology Department, “Sf. Apostol Andrei” Emergency County Hospital, 145 Tomis Blvd., 900591 Constanta, Romania; 7Urology Department, “Sf. Apostol Andrei” Emergency County Hospital, 145 Tomis Blvd., 900591 Constanta, Romania

**Keywords:** prostate carcinogenesis, cell cycle, apoptosis, CD34, CD61, CD42b glycoproteins, T cell infiltrations, Ki-67 expression

## Abstract

Prostate intratumoral heterogeneity, driven by epithelial–mesenchymal plasticity, contributes to the limited treatment response, and it is therefore necessary to use the biomarkers to improve patient prognostic survival. We aimed to characterize the tumor microenvironment (T lymphocyte infiltration, intratumoral CD34, and KI-67 expressions) by immunohistochemistry methods and to study the biological mechanisms (cell cycle, cell proliferation by adhesion glycoproteins, cell apoptosis) involved in the evolution of the prostate tumor process by flow-cytometry techniques. Our results showed that proliferative activity (S-phase) revealed statistically significant lower values of prostate adenocarcinoma (PCa) and benign prostatic hyperplasia (BPH) reported at non-malignant adjacent cell samples (PCa 4.32 ± 4.91; BPH 2.35 ± 1.37 vs. C 10.23 ± 0.43, *p* < 0.01). Furthermore, 68% of BPH cases and 88% of patients with PCa had aneuploidy. Statistically increased values of cell proliferation (CD34+ CD61+) were observed in prostate adenocarcinoma and hyperplasia cases reported to non-malignant adjacent cell samples (PCa 28.79 ± 10.14; BPH 40.65 ± 11.88 vs. C 16.15 ± 2.58, *p <* 0.05). The CD42b+ cell population with a role in cell adhesion, and metastasis had a significantly increased value in PCa cases (38.39 ± 11.23) reported to controls (C 26.24 ± 0.62, *p* < 0.01). The intratumoral expression of CD34 showed a significantly increased pattern of PCa tissue samples reported to controls (PCa 26.12 ± 6.84 vs. C 1.50 ± 0.70, *p* < 0.01). Flow cytometric analysis of the cell cycle, apoptosis, and adhesion glycoproteins with a critical role in tumoral cell proliferation, T cell infiltrations, Ki-67, and CD 34 expressions by IHC methods are recommended as techniques for the efficient means of measurement for adenocarcinoma and hyperplasia prostate tissue samples and should be explored in the future.

## 1. Introduction

Prostate adenocarcinoma (PCa) is the most common cancer in men in the world, being the leading cause of death. Despite its high incidence, PCa prognosis for patients is good when the carcinoma is detected in stages when androgen deprivation, prostatectomy, or/and radiation therapies are implemented [1,2,3]. Prostate intratumoral heterogeneity, driven by epithelial–mesenchymal plasticity, contributes to the limited treatment response; therefore, it is necessary to use the biomarkers to highlight this efficiently and quickly to improve patient prognostic survival. Flow cytometric analysis of ploidy and the cell cycle, together with adhesion glycoproteins with an essential role in tumoral cell proliferation and cell apoptosis, represent the rapid and efficient means of measurement for the microenvironment (TME) of the PCa and benign prostatic hyperplasia (BPH). DNA content observes the cell frequency in the G0/G1, S, and G2/M phases of the cell cycle and assesses DNA ploidy. The evidence of aneuploidy represents a marker of the tumor presence, being a prognostic indicator of tumor progression and the treatment outcome [4]. DNA ploidy and cell proliferation have provided prognostic information for prostate cancer [5]. Aneuploidy represents a human cancer characteristic, being a tumorigenesis driver [6]. Aneuploidy may arise during tumor initiation via polyploidization because unstable tetraploid intermediates determinate chromosomal gains, losses, and translocations [7,8,9]. Polyploid cells also occur in cancer, but aneuploidy cells are found in various tumors, often indicating higher malignancy. Flow cytometric analysis of aneuploidy has been used as a prognostic indicator in prostate, colon, and breast tumors, and it highlights that aneuploidy results from deletion or replication of specific chromosomes, with a different process from normal chromosome replication [10]. Cell adhesion molecules (glycoproteins) are essential in cancer progression and metastasis. Interactions between tumor cells, platelets, and leukocytes contribute to cancer cell adhesion, extravasation, and the establishment of metastatic lesions [11]. CD61 transmembrane glycoproteins (β3 integrin) attach the cells to the extracellular matrix (ECM), inducing cluster formation with signaling molecules (focal adhesions kinase), which result in cell adhesion and cell migration. Changes in integrin gene expression were shown in various malignancies, including prostate adenocarcinoma [12,13,14,15]. CD34 is a transmembrane phosphoglycoprotein associated with the proliferative capacity of multipotent mesenchymal stromal cells (MSC) [16,17]. In addition, CD34 represents a biomarker of vascular endothelial progenitor cells [18]. Noncirculating adult endothelial cells were represented by the CD34+ cell population, located within smaller blood vessels, while most endothelial cells from larger veins and arteries are part of the CD34- cell population. CD34+ endothelial cells are involved in migration and adhesion [17]. The integrin αIIb/β3 (CD61 complex) is involved in prostate cancer metastasis [19]. In addition, integrin α5β1 (GPIB-V complex) is vital in cell adhesion in prostate cancer cells [20]. T lymphocyte infiltration into the tumor microenvironment plays an important role in antitumor immunity [21]. T lymphocyte infiltration into malignant tumors in controlling cancer progression and survival of patients with cancer was described in [22]. Other authors suggest that patients with tumors with increased T lymphocyte infiltration have a survival advantage, but it appears that the mechanisms that contributed to escape the tumor cells originated from immune responses [3,23,24].

Our study presents the DNA content and cell apoptosis related to adhesion glycoprotein expressions and T lymphocyte infiltration in PCa and BPH tissue in a report with non-malignant adjacent tissue samples. DNA content was measured by flow cytometry to show cell distribution within the G0/G1, S, and G2/M phases of the cell cycle, to estimate the frequency of apoptotic cells with the fractional DNA content (subG0/G1), and to calculate the DNA ploidy of the observed cell population, being made by a PI stain. Adhesion glycoproteins made by dual stain CD34 Alexa Fluor 488 and CD61-PE reveal mesenchymal and endothelial cell proliferation and platelet and T cell aggregation to tumoral and endothelial cells (CD34+ CD61+). The CD42b-PE stain observed the platelet aggregation in tumor and endothelial cells. T lymphocytes, especially CD3 (total T lymphocytes), CD4 (helper T lymphocytes), and CD8 (cytotoxic T lymphocytes), were analyzed by immunohistochemistry methods (IHC) to observe the infiltration degrees of leucocytes in tissue samples. In addition, the intratumoral expression of CD34 and cell proliferation by Ki-67 expression in PCa and BPH tissue samples were reported to controls by ICH analysis.

## 2. Materials and Methods

### 2.1. Cases Selection

All tissue samples (*n* = 75) were recovered from patients (who signed informed consent forms, agreeing to participate in this study) from the Clinical Service of Pathology, Sf. Apostol Andrei Clinical Emergency County Hospital in Constanta, Romania. In agreement with WHO classifications, the patients were divided into two experimental groups: (1) patients with PCa without treatment (*n* = 25); (2) patients with BPH without treatment (*n* = 25); (3) controls for experimental groups using non-malignant adjacent tissue samples recovered from patients with PCa or BPH (*n* = 25, C, controls).

Tissue samples of PCa, BPH, and control, excised by transurethral resection of the prostate (TURP), were divided into two parts: (1) samples used to evaluate the T cell infiltrations, CD 34, and Ki-67 cell proliferation by IHC methods at Clinical Service of Pathology, Sf. Apostol Andrei Clinical Emergency County Hospital, Constanta, Romania; (2) samples mechanically homogenized with TissueRuptor II (Qiagen, USA), used for flow cytometry determinations (DNA content, cell apoptosis, CD34, CD61, and CD42b biomarkers at the Cell Biology Department, CEDMOG, Ovidius University of Constanta, Romania). Our selection criteria were applied to identify and establish the clinical efficiency of the human PCa and BPH biomarkers by flow cytometry and IHC methods, highlighting the characterization of the tumor microenvironment in conformity with references [25]. 

### 2.2. Morphological Evaluation of Tissue Samples

After the macroscopic description, the prostate tissue specimens were fixed in 10% formaldehyde, paraffin-embedded, sectioned, and stained in the usual laboratory stains. For the microscopic evaluation, by the Gleason classification, primary prostate adenocarcinomas were divided into three categories: (1) well-differentiated—Gleason score (GS) 6; (2) moderately differentiated—Gleason score 7; (3) poorly differentiated—Gleason scores 8–10. The second classification of PCa cases in the function of the prognostic grade of the patient survival was made in accordance with the references: (1) Group I—Gleason score ≤ 6 (*n* = 1); (2) Group II—Gleason score 3 + 4 = 7 (*n* = 16); (3) Group III—Gleason score 4 + 3 = 7; (4) Group IV—Gleason score 4 + 4 = 8 (*n* = 3); (5) Group V—Gleason score 9–10 (*n* = 5) [26,27,28]. After the T stage (pTNM), the third classification identified two risk groups: (1) patients with T1-T2 stage; (2) patients with T3-T4 stage [29].

### 2.3. Reagents and Equipment

Our study analyses used a flow cytometer (Attune, Acoustic focusing cytometer, Applied Biosystems, part of Life Technologies, Bedford, MA, USA). The flow cytometer was set using fluorescent beads (Attune performance tracking beads, labeling, and detection, Life Technologies, Europe BV, Bleiswijk, The Netherlands) with standard size (four intensity levels of beads population). The quantity was established by enumerating cells below 1 µm; 10,000 cells per sample for each analysis were gated by Forward Scatter (FSC) and Side Scatter (SSC). Flow cytometry data were collected using Attune Cytometric Software v.1.2.5, Applied Biosystems, 2010. Annexin V-FITC/PI (Bender MedSystems GmbH, Wien, Austria) was used to observe the apoptotic cells. Propidium iodide (PI) (1.0 mg/mL, Sigma-Aldrich, Chemie GmbH, Taufkirchen, Germany) and RNase A (4 mg/mL, Promega, Madison, WI, USA) were used in cell cycle analysis. Anti-CD42b-PE (HIP1) and anti-CD61-PE (integrin beta 3, Invitrogen, eBioscience) monoclonal antibodies conjugated with phycoerythrin (PE) were used to assess platelet glycoproteins expressions of GPIba and GPIIIa. CD34 Antibody, conjugate Alexa Fluor 488 (4H11(APG), Thermo Scientific, Waltham, MA, USA) was used for glycoprotein expression of CD34. Anti-CD4 (EP204 clone), anti-CD 8 (SP16 clone), anti-CD3 (EP41 clone), anti-CD34 (QB-End/10 clone), and anti-Ki67 (SP6 clone, Master Diagnostica, Sao Paolo, Brazil) monoclonal antibodies were used for immunohistochemistry (IHC) methods to evaluate the T cell infiltration, intratumoral expression of CD34, and cell proliferation in PCa and BPH tissue samples reported to controls. We used formalin-fixed, paraffin-embedded tissue for IHC assessment, sectioned at 4 µm thickness. We followed the staining protocol, as recommended by the producers. Master Diagnostica protocols included dewaxing using xylene and decreasing grades of alcohol, HIER in ph8 Master Diagnostica EDTA buffer in a pressure cooker incubating the ready-to-use monoclonal antibodies at room temperature for 10 min. For detection, we used the Master Polymer Plus Detection System (HRP) (DAB included), counterstained with hematoxylin, and mounted the glass cover slides.

### 2.4. Cell Cycle Analysis

In the darkness, the homogenized cells (100 µL) were introduced into flow cytometry tubes and fixed with 100 µL ethanol for 30 min. After this process, the cells were treated with 20 µL of PI (20 mg/mL) and 30 µL of RNase A (30 mg/mL) and incubated for 30 min at room temperature into darkness. Then, 1 mL flow cytometry stain buffer (FCB, eBioscienceTM, Life Technologies Europe BV, Bleiswijk, The Netherlands) was added, and the cell cycle distribution was detected with the flow cytometer using a 488 nm excitation and orange emission for PI (BL2 channel).

### 2.5. Adhesion Glycoproteins Determinations 

The homogenized cells (100 µL for each tube) spread: (1) CD61-PE and CD34+ -Alexa Flour 488 dual stain; (2) CD42b-PE stain; (3) control negative-IgG stain. In the tubes with cells were introduced 5µL of CD61-PE and 5µL of CD34-Alexa Flour 488. In the other tubes with cells were added 5 µL of CD42b-PE. A control tube with cells and 5 µL of the negative control (mouse IgG) were realized for each experimental sample. All work tubes were vortexed and incubated into darkness for 25 min at 37 °C. Then, 1 ml of FCB was added into each tube and vortexed for 1 min before analysis. Flow cytometry identified adhesion glycoproteins based on the size and specificity of CD34, CD61, and CD42b expressions, using the BL1 channel for Alexa Fluor 488 and the BL2 channel for PE.

### 2.6. Cell Apoptosis Assay

The homogenized cells from each sample were introduced in flow cytometry tubes with 2 µL of Annexin V-FITC and 2 µL PI (20 mg/mL) for 30 min at room temperature in darkness. After incubation, 1 mL of FCB was added. Viable cells, early apoptotic cells, late apoptotic cells, and necrotic cells were examined with a flow cytometer using 488 nm excitation, green emission for Annexin V-FITC (BL1 channel), and orange emission for PI (BL2 channel).

### 2.7. Surface Glycoproteins of T Cell Analysis

The expressions of the CD4, CD3, and CD8 biomarkers were evaluated semi-quantitatively as the number of lymphocytes (less or greater than 50 cells) on 400× magnification by microscope examination. Immunolabeling was considered brown positive at the membrane level. Lymphocytes from the peritumoral stroma were evaluated, either as single cells or as cell aggregates and intraepithelial cells. Tonsils were used as a positive control. Thus, the degree of inflammation in the tissue samples was assessed and divided into three categories depending on the lymphocyte’s percentage: (1) slight inflammation, when the inflammatory infiltrates were equal to or less than 10% (*n* = 39); (2) moderate inflammation, with infiltrates between 10 and 20 (*n* = 14); (3) severe inflammation, when the percentage of infiltrates was equal to or greater than 20 (*n* = 22) [30].

### 2.8. Intratumoral Expression of CD34

Vascularity was evaluated by an average of CD34-positive numbers of stained vessels in cases with non-malignant tissue, HBP tissue, and PCa tissue samples. CD34 expression was assessed in both small vessels within the prostate tumor tissue, which is more likely to be formed during tumor angiogenesis, and in pre-existing stromal vessels. For evaluation, the expression was observed by brown positivity in the monolayer endothelial cells, which line the lumen of small vessels. Each section was initially examined with a low power field of 100× to mark the area of MDV (microvessel density) and then with a large, high power field of 400×. They were evaluated as vascular spaces, lumen-centered structures, endothelial cell groups, or CD34-positive cell clusters that were considered a microvessel. In the situation where at least two clusters or foci were observed, which seemed to belong to the same vascular structure, they were also considered a microvessel. MVD count was established as the sum of the three highest counts, in the hot spots, at 0.18 mm^2^ [29,31]. 

### 2.9. Ki-67 Cell Proliferation

The evaluation was quantified at the nuclear level by strong brown staining as the percentage of cells that react with the antibody. Positive cells were counted from 500 cells evaluated on the magnification of 400 (HPF—high power field). We used as reference for interpretation the following values: below 2%, negative (*n* = 50); below 25%, score 1+ (*n* = 25); 26–50%, score 2+; 51–75%, score 3+; 76–100%, score 4+ [32].

### 2.10. Statistical Analysis

We analyzed the cell cycle, adhesion glycoproteins, cell apoptosis, T cell infiltrations, and intratumoral expression of CD34 for all tissue samples, and the obtained results were presented as mean values with standard deviations, made by SPSS v. 23 software, IBM, Armonk, NY, USA, 2015. Data were analyzed by the Levene test for homogeneity of the sample variances, an independent *t* test was used to show the differences between cases, and *p* < 0.05 was considered statistically significant. The Pearson correlations were made between DNA content, cell apoptosis, glycoproteins parameters, tissue inflammation grade, and Ki-67 cell proliferation in PCa and HP tissue in a report with non-malignant adjacent tissue samples. Figure 1, Figure 2, Figure 3 and Figure 4 were produced with Attune Cytometric Software v.1.2.5, Applied Biosystems, Bedford, MA, USA, 2010.

## 3. Results

### 3.1. DNA Content, CD34, CD61, CD42 b Glycoproteins Expressions, Cell Apoptosis by Flow Cytometry Analysis in PCa and BPH Tissue Reported to Non-Malignant Adjacent Tissue Samples 

The proliferative activity (S-phase) revealed statistically significant lower values of prostate adenocarcinoma and hyperplasia reported at non-malignant adjacent cell samples (PCa 4.32 ± 4.91; BPH 2.35 ± 1.37 vs. C 10.23 ± 0.43, *p* < 0.01, Table 1, Figure 1). Furthermore, 68% of BPH cases presented aneuploidy, 88% of patients with prostate adenocarcinoma had aneuploidy, 4% of cases from these highlighted hypodiploid and hyperdiploid cell heterogeneity, and 4% of cases showed polyploidy (tetrapoliploidy). 

Index of ploidy presented significantly increased values at PCa and BPH cases reported to control cases (PCa 1.38 ± 0.51; BPH 1.19 ± 0.20 vs. C 1.00 ± 0.01, *p* < 0.01). DNA fragmentation (subG0/G1) had a significantly increased percentage in PCa tissue (4.63 ± 2.69) compared to control tissue samples (1.43 ± 0.11, *p* < 0.01, Table 1, Figure 1). 

Statistically increased values of cell proliferation (CD34+ CD61+) were observed in prostate adenocarcinoma and hyperplasia cases reported to non-malignant adjacent cell samples (PCa 28.79 ± 10.14; PH-40.65 ± 11.88 vs. C 16.15 ± 2.58, *p* < 0.05, Table 2, Figure 2). 

CD 61+ cell population, characteristically for platelets and T cell aggregation to tumoral cells and endothelium, presented significantly increased levels of PCa (37.81 ± 16.21) and BPH (36.50 ± 13.55) samples reported in controls (16.26 ± 2.52, *p* < 0.01). Mesenchymal cell proliferation represented by the CD34+ cell population revealed a significant increase in BPH cases reported to controls (66.31 ± 11.28 vs. 54.49 ± 0.72, *p* < 0.01). In addition, benign prostatic hyperplasia showed a significantly lower pattern for the CD34- cell population reported to control cases (32.69 ± 11.30 vs. 45.25 ± 0.52, *p* < 0.01, Table 2), which characterized the tumoral cells’ adhesion to endothelial cells from larger veins and arteries.

A significant increase in the CD42b+ cell population with a role in cell adhesion and metastasis was observed in PCa cases (38.39 ± 11.23) reported to controls (26.24 ± 0.62, *p* < 0.01). Instead, the CD42b- cell population presented significantly lower values in PCa and BPH tissue samples compared to non-malignant adjacent cell samples (PCa 59.26 ± 12.02; BPH 59.97 ± 17.15 vs. C 73.14 ± 0.08; *p* < 0.01; *p* < 0.05, Table 3, Figure 3).

Cell viability had increased values without significant differences on PCa and control tissue samples (81.30 ± 16.42; 86.95 ± 5.75, *p* > 0.05, Table 4, Figure 4), but the obtained results for BPH cases showed a significantly lower value of this on BPH tissue samples reported to controls (BPH 64.26 ± 22.68 vs. 86.95 ± 5.75, *p* < 0.05). Necrosis presented a statistical increase of values in BPH cases compared with control cases (BPH 26.76 ± 6.32 vs. C 13.04 ± 5.76), and incipient cell apoptosis had a statistical increase in PCa tissue samples (7.08 ± 10.46) reported to non-malignant adjacent tissue samples (0.10 ± 0.01, *p* < 0.05, Table 4, Figure 4).

### 3.2. Surface Glycoproteins of T Cells, Intratumoral Expression of CD34, and Ki-67 Cell Proliferation by ICH Analysis in PCa and BPH Tissue Reported to Non-Malignant Adjacent Tissue Samples 

This study presented prostate tissue samples with T cell infiltrates from areas of PCa and BPH versus non-malignant adjacent tissue regions. PCa samples presented similar patterns for CD3+, CD4+, and CD8 lymphocytes that formed clusters adjacent to adenocarcinoma areas, which appeared separated from the lymphocytic infiltration. Healthy prostate tissue samples contain CD3+, CD4+, and CD8+ lymphocytes dispersed in the interstitial stroma without cluster formation. BPH tissue samples presented CD3+, CD4+, and CD8 lymphocytes with their distribution, such as healthy tissue, but they sometimes may form significantly smaller clusters than those of the adenocarcinoma tissue. The total CD3 lymphocytes presented slightly higher patterns without significant differences in PCa and BPH reported in non-malignant tissue samples (59.00 ± 22.43, 57.70 ± 25.48 vs. 44.00 ± 22.62, *p* > 0.05). The cytotoxic CD8 lymphocytes had slightly lower PCa and BHP tissue sample values without significant differences in controls (PCa 31.25 ± 17.55 and BPH 26.00 ± 14.49 vs. C 45.00 ± 28.28, *p* > 0.05, Table 5, Figure 5). 

The intratumoral expression of CD34 showed a significantly increased pattern of PCa tissue samples reported to controls (26.12 ± 6.84 vs. 1.50 ± 0.70, *p* < 0.01, Table 5, Figure 6). In addition, Ki-67 was expressed in only PCa tissue samples, the score of cell proliferation being below 25% (score 1+) for all studied PCa cases (Figure 6).

### 3.3. Correlations between DNA Content, Cell Apoptosis, Adhesion Glycoproteins Expressions, Tissue Inflammation Grade, and Ki-67 Cell Proliferation in Tissue Samples

Relationships between the DNA content and adhesion glycoproteins are presented in Table 6. Pearson correlations were observed between the G0/G1 phase of cell cycle and CD34+ CD61+ glycoproteins (r = −0.514; *p* < 0.01), and CD42b+ cell population (r = −0.475; *p* < 0.05), between G2/M phase of cell cycle and CD34+ CD61+ glycoproteins (r = 0.513; *p* < 0.01), and CD42b+ cell population (r = 0.446; *p* < 0.05), between S phase and CD61+ cell population (r = −0.430; *p* < 0.05, Table 6). 

Double-positive population of glycoproteins (CD34+ CD61+) were positively correlated with total CD61 and CD34 glycoprotein levels (r = 0.512; r = 0.503, *p* < 0.01), CD61+, CD34+, and CD42b+ cells populations (r = 0.589; r = 0.500; r = 0.623; *p* < 0.01), and were negatively correlated with CD61-, CD34-, and CD42b- cell populations (r = −0.550; r = −0.516; r = −0.641; *p* < 0.01, Table 7).

The grade of tissue inflammation by T lymphocytes was directly correlated with CD 61, CD42b-positive cell populations (r = 0.544; *p* < 0.05; r = 0.664, *p* < 0.01), late apoptosis (r = 0.528; *p* < 0.05) and was inversely correlated with CD 61, CD42b-negative cell populations (r = −0.535; *p* < 0.05; r = −0.639, *p* < 0.01), and cell viability (r = −0.632; *p* < 0.01, Table 8).

Cell proliferation by Ki-67 expression was positively correlated with cell viability (r = 0.548; *p* < 0.05) and intratumoral CD34 expression (r = 0.611; *p* < 0.01), and was negatively correlated with necrosis (r = −0.682; *p* < 0.01, Table 9).

## 4. Discussion

The cell cycle is characterized by the interphase and mitosis phases. Interphase is represented by three sub-phases, G1, S, and G2. In G1, cells based on internal/external signals lead to a decision of DNA replication or not [4]. The S phase is defined by the ability to synthesize genomic DNA. G2 is the second gap between S and Mitosis, with a function such as DNA damage repair and preparation for entering into Mitosis (M phase). G0/G1, S, and G2/M phases of the cell cycle are quantitatively identified by the flow cytometry method based on propidium iodide stain and RNase. 

Ploidy and cell cycle analyses were the first flow cytometry applications, being rapid and efficient measurement methods [33,34]. DNA ploidy is defined as DNA index (DI), and for normal cells in the G0/G1 phase of the cell cycle, DI is 1.0. Aneuploid/polyploid cell populations are divided by DI distribution into categories such as hypodiploid (DI < 0.95), hyperdiploid (DI = 1.15−1.91), tetraploid (DI = 1.92−2.04), hypertetraploid (DI ≥ 2.05), and multiploid (DNA content histogram has ≥2 peaks corresponding to aneuploid/polyploid cell population) [35]. Apoptotic cell frequency that is characterized by fractional DNA content is defined as a subG0/G1 cell population [4].

In our study, 68% of BPH cases present aneuploidy. In addition, 88% of PCa patients had aneuploidy, including hypodiploid and hyperdiploid cell heterogeneity and tetrapoliploidy. PCa samples presented modified cell cycle phases, represented by cell cycle arrest in G0/G1 and a low S phase. 

The authors proposed to gain prognostic information by dividing the S proliferative phase (S-phase) into three prognostic categories: low (<7.0%), intermediate (7.0−11.9%), and high (≥12%). These categories allow for the grouping of the patients according to their level of risk. The risk of death or recurrence for diploid and aneuploid cases is 50% higher for the high S-phase category, and 50% higher for the intermediate category than for the low category. Despite different techniques (the tissue samples are fresh, frozen, or paraffin-embedded), a higher S phase of the cell cycle is correlated with worse tumor grade and larger tumors in breast cancer tissue samples [35]. Lower values of the S-phase for PCa and BPH cases were observed in our study, included in the first category of prognostics reported by the references. In addition, we observed a significant negative correlation between the S-phase and CD 61 cell positive population, which means a better survival rate for the patients. 

Adhesion glycoproteins from this study, represented by the increased values of double-positive populations of CD34/CD61, and the platelets, T cell aggregation to tumoral cells and endothelium, represented by the CD61+ cell population, observed at PCa and BPH cases, support adhesion, migration, and cell proliferation and are in accord with the following references.

Integrins, the transmembrane glycoprotein receptor superfamily, are represented in our study by the CD61 and CD42b glycoproteins with a role in cancer progression. Altered cell adhesion leads to cell proliferation, migration, and metastasis correlated to the different stages of human tumors and pathological outcomes (metastasis, recurrence, survival) [36,37,38,39,40,41,42,43,44].

Other integrins, with roles in tumoral cell adhesion and metastasis, such as the CD42b+ cell population had increased values in PCa cases, conforming to the following references. β integrins are transmembrane protein receptors that attach cells to the ECM or bind ligands secreted by other cells. The type I membrane glycoproteins (CD42b) play essential roles in cell signaling networks, growth, differentiation, mobility, and survival [12,15,45].

Integrins are essential in acquiring and maintaining the neoplastic phenotype by escaping from cell apoptosis and maintaining cell proliferation. Integrin expression is modified upon the normal-to-neoplastic transition. Activated platelet integrins αIIbβ3 help the cancer cells of blood circulating from induced tumor cell arrest by binding to leukocytes and platelets to survive a long time [46]. 

α1β1, α2β1, and α5β1 integrin activation stimulates vascular endothelial growth factor (VEGF) expression, promotes VEGF receptor activation, and increases the adhesion of endothelial cells to ligands (angiogenic effect) [11,47,48]. 

Other authors observed that cell platelet interactions are mediated either by *p*-selectin or platelet integrin αIIbβ3 in metastasis. αIIbβ3 integrins or *p*-selectin inhibition by function-blocking antibodies determine lower platelet-tumor cell interaction and tumor cell adhesion on activated endothelium [11]. A bridging factor between platelet αIIbβ3 integrins and tumoral cells was identified as a fibrinogen that facilitates tumor cell arrest in the vasculature and metastasis to various tissues [46]. The authors investigated the role of β1 integrins in tumor growth and metastasis and observed that β integrin overexpression correlated with the metastatic spread of these cells to the lung and liver [11].

Metastasis is the common cause of cancer-related deaths because it is based on the complex formation process of the migratory cells, named the epithelial–mesenchymal transition (EMT). The cell-to-cell and cell-to-matrix adhesion molecule expressions are essential to metastasis formation. The leukocytes/cancer cells’ attachment to the endothelium is mediated by different integrins [49]. L1-CAM ligands interact with integrins such as α5ß1, αVß5, αVß1, αVß3, and αIIbß3 with a role in the adhesion process in tumor cell extravasation [50]. In agreement with the references presented above, we observed that CD61+/CD34+ glycoproteins were positive and significantly correlated with the biomarker of cell metastasis (CD42b+). In BPH cases, mesenchymal cell proliferation is represented by higher values of the CD34+ cell population. According to the references, the characterization of the tumoral cells’ adhesion to endothelial cells from larger veins and arteries is represented by a lower pattern for the CD34- cell population.

Mesenchymal cells (MSC) present in *in vitro* mesenchymal differentiation potential, which is well reported by the references. These are associated with properties such as paracrine wound healing, niche forming abilities, immune privilege, and immunomodulation [51,52]. Is was reported that the freshly extracted stromal cells from various tissues contain a CD34+ cells population with distinct characteristics from the total MSC population [53]. CD34+ cells were associated with MSC biomarkers such as CD271 and Stro-1 and biomarkers such as CD45 and CD133 [16,53,54,55,56]. CD34+ cells have a greater tendency for endothelial transdifferentiation [57,58]. CD34 was found on embryonic stem cell-derived MSC, suggesting that it is a marker of early human MSC [59]. In addition, CD34 expression was observed on the luminal membrane of cellular processes and the abluminal membrane of cells found at the tips of vascular sprouts [15]. 

PCa has a lower proliferative capacity, which renders apoptosis induction important for targeted therapies, especially for studying the apoptotic signaling mechanisms responsible for apoptosis evasion [60]. In our study, cell viability was increased for PCa samples, without significant differences reported to the controls. An easily increased but significant value of incipient cell apoptosis was observed in PCa cases. Instead, BPH cases presented a lower pattern of cell viability and a higher necrosis value reported to controls. Our observations are essential to understanding the molecular mechanisms implied in cell apoptosis in PCa and BPH cases because they seem to be different. Cancer progression results from an imbalance in cell proliferation and apoptosis. 

Anoikis represents a form of cell apoptosis based on the detachment of cells from the extracellular matrix (ECM), which is evaded by tumor cells to spread [60,61]. This is a developed strategy by the tumor cells in the metastatic spread and therapeutic resistance. Tumor cells undergoing EMT can evade the anoikis based on cellular reprogramming. Pro-EMT molecules such as transcriptional repressors SNAIL and SLUG and cell adhesion molecules confer the resistance of tumoral cells to anoikis [62].

Prostate cancer cells can modify their integrin expressions to lead to an anoikis-resistant phenotype. Integrins in the prostate and other cancers confer a migratory phenotype. Anoikis and EMT processes contribute to chemoresistance, immune evasion, and metastasis [61]. Biochemical recurrence of prostate cancer (bcr) and metastasis prediction after PCa curative treatment were shown by the references for other molecular tests [63].

Another objective of our study was to observe lymphocyte infiltrations, their activation status, and their distribution in PCa and BPH tissues compared to non-malignant prostate tissue samples. T cell infiltrations provide information about the immune system–tumoral cell interactions and the immune evasion mechanisms, which are necessary for developing anti-tumoral immunotherapies.

T cell infiltration in PCa samples forms the clusters adjacent to adenocarcinoma areas, separated from lymphocytic infiltration. The T lymphocytes are dispersed in the interstitial stroma in controls without cluster formation. In addition, T lymphocyte distribution in BPH cases is similar to non-malignant tissue. Analysis of lymphocyte distribution in the tumor environment was described for different tumors [22,64]. An intense infiltration of CD3+ cells is related to slow progression and better prognosis. Our study presented directly correlated T cell infiltrations and CD61+ and CD42b+ adhesion glycoproteins. We observed that an intense T cell infiltration contributes to the generally small tumor size of PCa. A similar association was observed in small-cell lung cancer, where a high number of T cells was associated with a significantly smaller tumor size [65]. The immune cells’ recruitment and interactions with the prostate microenvironment promote PCa progression. Studies about the profiles of the prostate tumor-infiltrating lymphocytes are limited. It has been reported that the immune response might have anti- or pro-tumorigenic potential, depending on cell phenotypes and the tumor microenvironment. Distribution and inflammatory cell interactions in the PCa and BPH represent promising indicators of the potential response of the target cell populations in immunotherapies and biomarkers to measure therapeutic efficacy [66,67]. The causal relationship between tumor proliferation and inflammation is widely studied, and in this regard, chronic inflammation has a vital role in developing malignant epithelial tumors [68]. A causal link between inflammation in normal prostate tissue and samples with a diagnosis of malignancy was also observed [66]. In this regard, tumor cells, including prostate malignant neoplastic cells, are modified, atypical cells that can induce a strong immune response in the body, which during the inflammatory process, are consumed, and the lymphocyte-mediated antitumor immune response is gradually reduced [69,70,71].

Intense tumor neovascularization is closely associated with tumor growth and metastasis. Angiogenesis is thereby a crucial factor affecting the prognosis of cancer patients. The analysis of microvessel density (MVD) by CD34, made in our study, showed a significant increase in the number of microvessels in PCa compared to BPH and non-malignant tissue samples. Intratumoral CD34 represents a biomarker for the IHC visualization of microvessels in benign and malignant prostate tissue. The authors observed that in PCa, a sensitive biomarker for newly derived blood vessels was CD34, and IHC analysis and MVD quantification within the tumor represents the basis for understanding the effects of antiangiogenic treatment [72]. CD34, a myeloid progenitor cell antigen also expressed by endothelial cells of arteries and venules, is considered the most sensitive and stable vascular marker, with a high positive rate and expression level [73]. In addition, the expression of CD34 in new vessels—tiny ones, compared to large ones, considered to be old vessels—in the tumor microscopic field suggests and strengthens the idea that this vascular marker plays a crucial role in the process of tumor neoangiogenesis [74]. Angiogenesis represents a prognostic factor by using CD34 as an endothelial biomarker in various solid tumors, including prostate adenocarcinoma [75,76,77]. A high immunoexpression of CD34 in the vessels of tumor tissue indicates intensive neoplastic neovascularization and increased MVD. According to some authors, increased MVD was associated with increased PSA and Gleason scores and later clinical stage, which may be due to rapid tumor growth induced by a high nutrient supply rate of newly formed blood vessels [29]. The authors observed a strong correlation between GS and therapeutic response to cabazitaxel in metastatic castration-resistant prostate cancer patients [78]. 

As men are diagnosed with prostate cancer at an older age than other malignancies, more attention should be paid to markers of anti-angiogenic activity. Combining anti-angiogenic drugs with other drugs of different classes may open a door for more promising clinical results [79].

Another studied biomarker, Ki-67 cell proliferation, was expressed only in prostate adenocarcinoma tissue samples, agreeing with authors who also observed that this biomarker is expressed in PCa reported in BPH tissue samples [29,80,81]. In addition, a significant positive correlation between Ki-67 staining and intratumoral CD34 expression was observed. It was reported that Ki 67 as a proliferating biomarker has higher accuracy in the early diagnosis of PCa [82], although qualifying it as an independent prognostic marker in prostate adenocarcinoma is still controversial. According to some authors, the Ki-67 index is more expressed in adenocarcinoma tissues than in benign prostate hyperplasia and is still higher in metastatic than non-metastatic cases. Thus, an increased Ki-67 value may indicate a poor prognosis of the disease [32]. Our study observed a direct correlation between Ki 67 cell proliferation and intratumoral CD34 expression, but no correlation between Ki67 expression and Gleason scores, as observed in the study [81], which concluded that this nuclear biomarker could be a prognostic factor for prostate cancer.

In this study, the principal limitation to developing the utility of cell cycle, apoptosis, and adhesion glycoproteins, T cell infiltrations, Ki-67, and CD 34 expressions by flow cytometry and IHC methods in adenocarcinoma and hyperplasia prostate cases was a small number of samples recovered from the patients. Another limitation of our study may be represented by the heterogeneity of the PCa tissue samples. With the macro-dissection technique, it is difficult to purify the tumor and non-malignant parts because some cancer parts may still have BPH tissue. For the ideal cases, micro-dissection with laser captures may be needed, but in Romania, only macro-dissection is used to separate the tumoral from the non-malignant parts of tissue samples made by experienced pathologists.

Future directions in this research area will be to study the importance of these biomarkers in many malignant affections because there is a promise to be critical regarding diagnostic biomarkers for diseases and uses to improve patient prognostic survival. In addition, these biomarkers may be studied in the prostate cell line co-culture system, which may provide more information about the biological mechanisms implied in the characterization of the tumoral microenvironment.

## 5. Conclusions

Biological mechanisms implied in the prostate tumor microenvironment characterization represented by the cell cycle, apoptosis, adhesion glycoproteins, T lymphocytes infiltrations, Ki-67, and intratumoral CD 34 biomarkers provide efficient means of measurement by flow cytometry and IHC techniques for PCa and BPH tissue samples and should be explored in the future not only for diagnostics but also for therapeutic purposes.

## Figures and Tables

**Figure 1 biomedicines-10-01672-f001:**
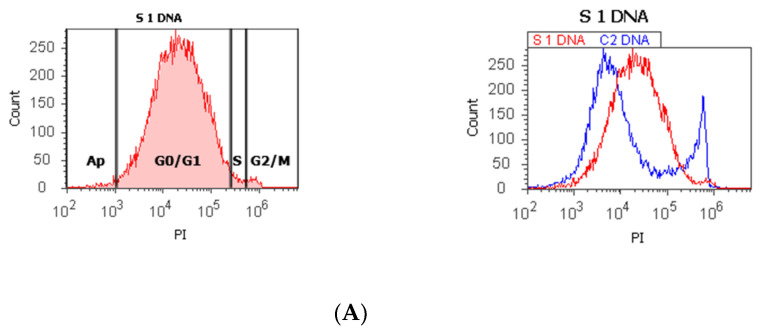
Cell cycle, ploidy, and DNA fragmentation (AP, subG0/G1) represented by propidium iodide stain (PI). Control DNA extrapolate on the PI ax. *G0/G1 phase* = (**A**) 95.48%; (**B**) 2c 59.44%; 4c 10.87%; 8c 6.52%; 16c 12.68%; (**C**) 0.00%; *S phase* = (**A**) 1.27%; (**B**) 0.00%; (**C**) 0.00%; *G2/M phase =* (**A**) 1.13%; (**B**) 0.00%; (**C**) 99.92%; *subG0/G1: **(*****A**) 1.05%; (**B**) 3.31%; (**C**) 0.07%. Legend: AP, cell apoptosis; (**A**) malignant tumor tissue with an invasion of the prostatic urethra (hyperdiploid, cell cycle arrest in G0/G1 phase); (**B**) adenoleiomyoma prostate tissue, chronic inflammation hyperplasia (tetrapoliploidy); (**C**) benign prostatic tissue hyperplasia (aneuploid, present only the G2/M phase).

**Figure 2 biomedicines-10-01672-f002:**
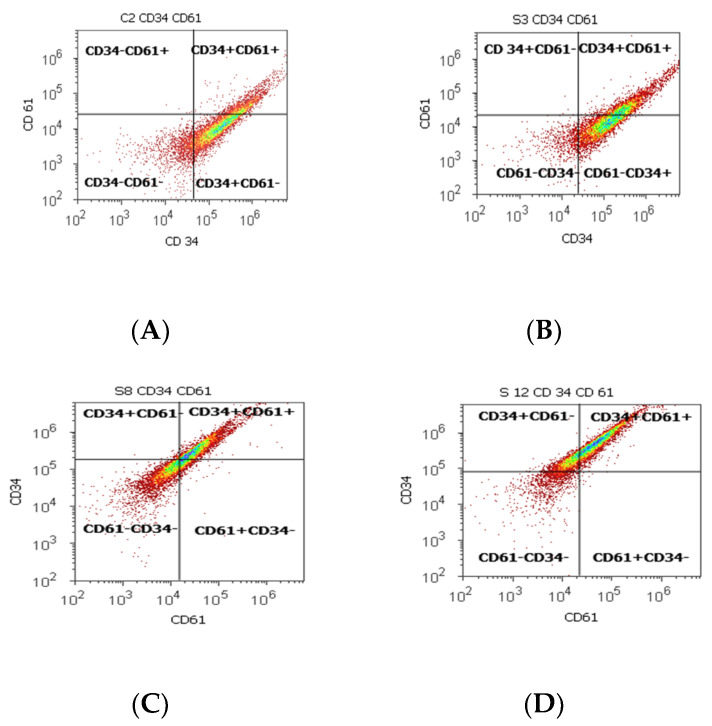
Cell proliferation highlights the double-positive populations of glycoprotein expression (CD34+ CD61+) with CD34- Alexa Fluor 488 and CD 61-PE dual stain. Cell adhesion is interpreted by the CD34 + CD61+ population: (**A**) 17.98%; (**B**) 33.63%; (**C**) 45.71%; (**D**) 59.68%. Legend: (**A**) non-malignant prostate tissue adjacent to nodular hyperplastic tissue; (**B**) adenoleiomyoma prostate tissue, chronic inflammation hyperplasia; (**C**) malignant prostate tumor tissue; (**D**) benign prostatic tissue hyperplasia.

**Figure 3 biomedicines-10-01672-f003:**
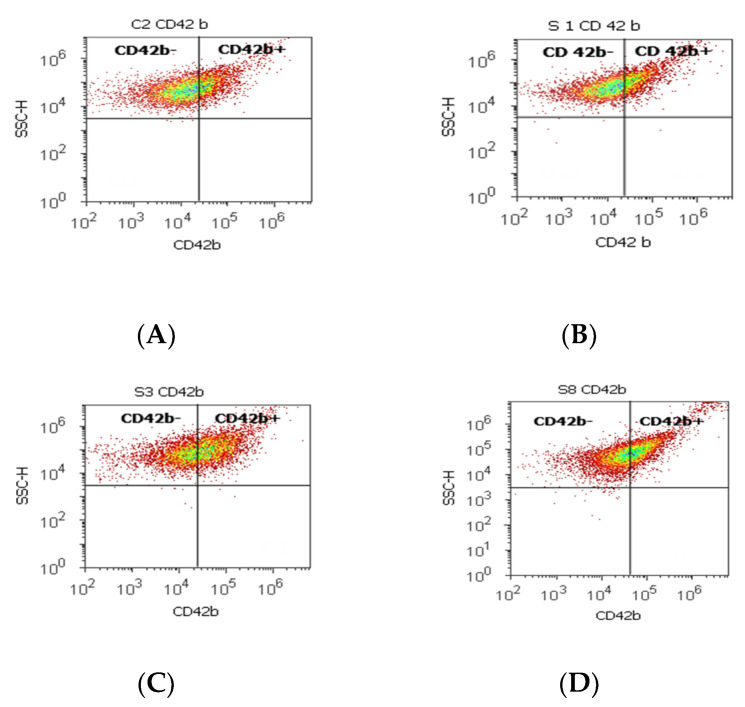
Integrin mediation of platelet aggregation to tumoral and endothelial cells by highlighting the positive and negative glycoprotein populations (CD42b-/CD42b+) with CD 42b-PE stain. CD42b+ population: (**A**) 25.80 %; (**B**) 16.79%; (**C**) 41.10%; (**D**) 39.74%; CD42b- population: (**A**) 73.08%; (**B**) 82.88%; (**C**) 57.58%; (**D**) 57.23%. Legend: (**A**) non-malignant prostate tissue adjacent to nodular hyperplastic tissue; (**B**) adenoleiomyoma prostate tissue, chronic inflammation hyperplasia; (**C**) malignant prostate tumor tissue; (**D**) benign prostatic tissue hyperplasia.

**Figure 4 biomedicines-10-01672-f004:**
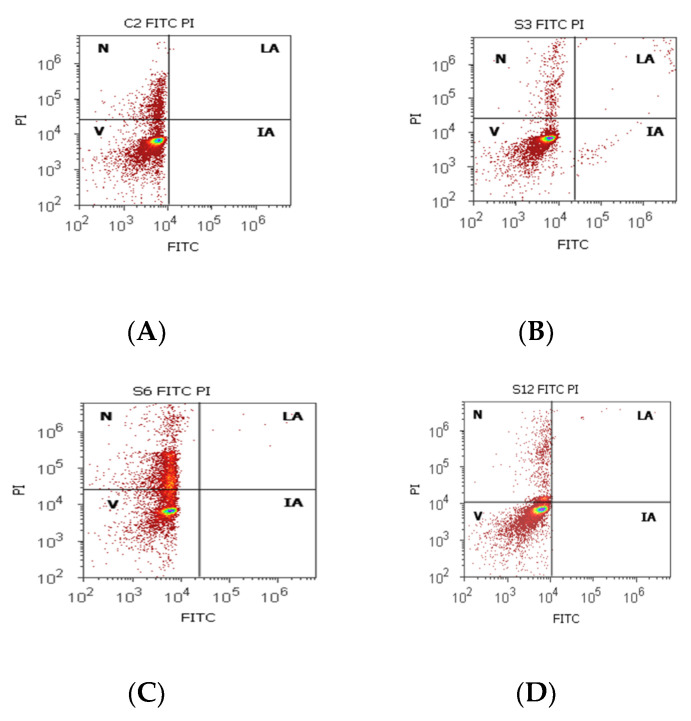
Cell apoptosis by Annexin V- FITC/ propidium iodide (PI) dual stain. Viability (V): (**A**) 83.50%; (**B**) 94.72%; (**C**) 68.79%; (**D**) 87.90%; Incipient apoptosis (IA): (**A**) 0.00%; (**B**) 0.54%; (**C**) 0.005%; (**D**) 0.025%; Late apoptosis (LA): (**A**) 0.00%; (**B**) 0.49%; (**C**) 0.07%; (**D**) 0.22%; Necrosis (*n*): (**A**) 17.12%; (**B**) 4.23%; (**C**) 31.13%; (**D**) 11.85%. Legend: A, non-malignant prostate tissue adjacent to nodular hyperplastic tissue; B, adenoleiomyoma prostate tissue, chronic inflammation hyperplasia; C, malignant prostate tumor tissue; D, benign prostatic tissue hyperplasia.

**Figure 5 biomedicines-10-01672-f005:**
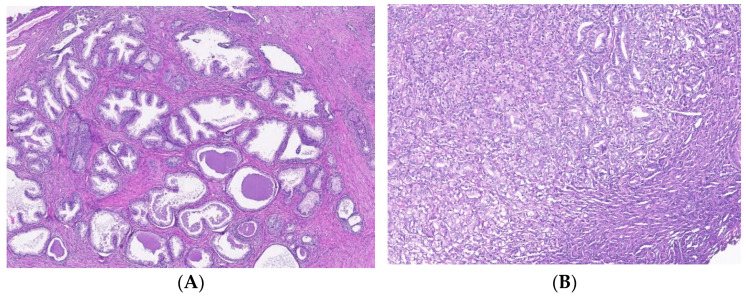
T cell infiltrates from areas of PCa and BPH tissue samples. Legend: (**A**) Microscopic appearance of benign prostatic hyperplasia, usual staining (×20); (**B**) prostate adenocarcinoma, usual staining, Gleason score 3 + 4 (×20); (**C**) CD4-positive peritumoral lymphocytes, more than 50 cells/HPF, prostate adenocarcinoma, Gleason score 3 + 4 (×20); (**D**) CD4-intensely positive in the peritumoral stroma, more than 50 cells/HPF, prostate adenocarcinoma, Gleason score 4 + 5 (×20); (**E**) CD3-positive (adenocarcinoma, Gleason score 4 + 5), less than 50 cells /HPF (×20); (**F**) CD8-positive (adenocarcinoma, Gleason score 4 + 4), more than 50 cells /HPF (×20).

**Figure 6 biomedicines-10-01672-f006:**
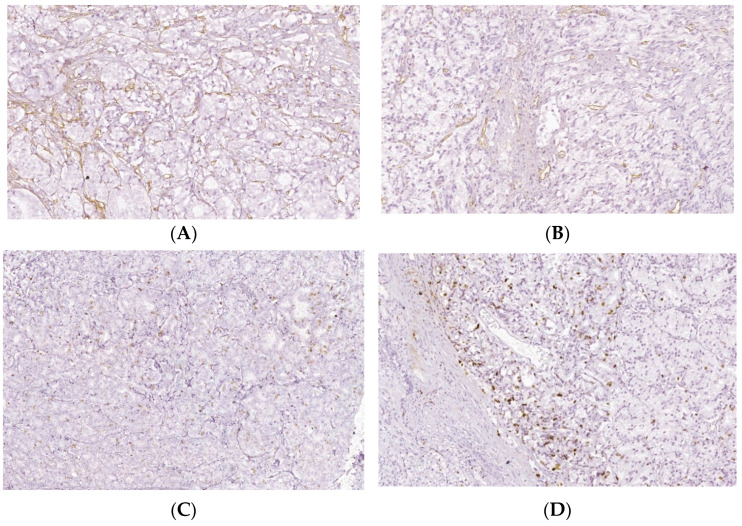
Intratumoral CD34 and Ki-67 expressions of PCa and BPH tissue samples. Legend: (**A**) CD34 positive vascular endothelial, intratumoral, 10–20 vessels/HPF, prostate adenocarcinoma, Gleason score 4 + 3 (×20); (**B**) CD34 positive intratumoral vasculature, 20–30 vessels/HPF adenocarcinoma, Gleason score 4 + 4 (×20); (**C**) Ki67 nuclear positive, score 1+, less than 25% of tumor cell proliferation 9 (×40); (**D**) Ki67 nuclear positive, score 1+, less than 25% of tumor cell proliferation (×40).

**Table 1 biomedicines-10-01672-t001:** Cell cycle phases, ploidy index, and DNA fragmentation at prostate adenocarcinoma, benign prostatic hyperplasia, and non-malignant adjacent tissue samples.

Number	Parameters	Control X ± SD	PCa X ± SD	Control X ± SD	BPH X ± SD
1	G0/G1 phase	68.67 ± 1.51	76.84 ± 20.68	68.67 ± 1.51	66.74 ± 34.38
	*p* values	0.14		0.86	
2	S phase	10.23 ± 0.43 **	4.32 ± 4.91 **	10.23 ± 0.43 **	2.35 ± 1.37 **
	*p* values	**0.00**		**0.00**	
3	G2/M phase	14.64 ± 2.99	8.81 ± 14.23	14.64 ± 2.99	20.42 ± 33.03
	*p* values	0.19		0.60	
4	DNA index	1.00 ± 0.01 *	1.38 ± 0.51 *	1.00 ± 0.01 *	1.19 ± 0.20 *
	*p* values	**0.01**		**0.03**	
5	subG0/G1	1.43 ± 0.11 *	4.63 ± 2.69 *	1.43 ± 0.11	2.19 ± 2.74
	*p* values	**0.01**		0.40	

Legend: X, obtained results mean; SD, standard deviation; PCa, prostate adenocarcinoma; BPH, benign prostatic hyperplasia; ** *p* ≤ 0.01 and * *p* < 0.05 represent statistically significant differences between controls and experimental samples made by independent *t* test.

**Table 2 biomedicines-10-01672-t002:** CD34 and CD61 glycoprotein expressions implied in cell proliferation in prostate adenocarcinoma, benign prostatic hyperplasia, and non-malignant adjacent tissue samples.

Number	Parameters	Control X ± SD	PCa X ± SD	Control X ± SD	BPH X ± SD
1	CD61T	86.90 ± 4.39 *	70.59 ± 15.30 *	86.90 ± 4.39	91.15 ± 4.76
	*p* values	**0.019**		0.273	
2	CD34 + CD61+	16.15 ± 2.58 **	28.79 ± 10.14 **	16.15 ± 2.58 *	40.65 ± 11.88 *
	*p* values	**0.005**		**0.019**	
3	CD61+	16.26 ± 2.52 **	37.81 ± 16.21 **	16.26 ± 2.52 **	36.50 ± 13.55 **
	*p* values	**0.000**		**0.002**	
4	CD61-	83.49 ± 2.72 **	59.89 ± 17.38 **	83.49 ± 2.72 **	62.49 ± 13.57 **
	*p* values	**0.000**		**0.001**	
5	CD34T	87.29 ± 5.20	75.43 ± 14.20	87.29 ± 5.20	90.55 ± 6.38
	*p* values	0.090		0.532	
6	CD34+	54.49 ± 0.72	57.81 ± 14.22	54.49 ± 0.72 **	66.31 ± 11.28 **
	*p* values	0.369		**0.009**	
7	CD34-	45.25 ± 0.52	39.95 ± 13.36	45.25 ± 0.52 **	32.69 ± 11.30 **
	*p* values	0.136		**0.007**	

Legend: X, obtained results mean; SD, standard deviation; PCa, prostate adenocarcinoma; BPH, benign prostatic hyperplasia; CD61 T, total CD61 glycoproteins expression; CD34 T, total CD34 glycoproteins expression; ** *p* ≤ 0.01 and * *p* < 0.05 represent statistically significant differences between controls and experimental samples made by independent *t* test.

**Table 3 biomedicines-10-01672-t003:** CD42b glycoprotein expression at prostate adenocarcinoma, benign prostatic hyperplasia, and non-malignant adjacent tissue samples.

Number	Parameters	Control X ± SD	PCa X ± SD	Control X ± SD	BPH X ± SD
1	CD42bT	86.98 ± 5.23	76.07 ± 12.95	86.98 ± 5.23	77.75 ± 17.37
	*p* values	0.113		0.209	
2	CD42b+	26.24 ± 0.62 **	38.39 ± 11.23 **	26.24 ± 0.62	38.06 ± 17.43
	*p* values	**0.001**		0.061	
3	CD42b-	73.14 ± 0.08 **	59.26 ± 12.02 **	73.14 ± 0.08 *	59.97 ± 17.15 *
	*p* values	**0.000**		**0.038**	

Legend: X, obtained results mean; SD, standard deviation; PCa, prostate adenocarcinoma; BPH, benign prostatic hyperplasia; CD42b T, total CD42b glycoprotein expression; ** *p* ≤ 0.01 and * *p* < 0.05 represent statistically significant differences between controls and experimental samples made by independent *t* test.

**Table 4 biomedicines-10-01672-t004:** Cell apoptosis at prostate adenocarcinoma and benign prostatic hyperplasia cases reported to control cases.

Number	Parameters	Control X ± SD	PCa X ± SD	Control X ± SD	BPH X ± SD
1	Viability	86.95 ± 5.75	81.30 ± 16.42	86.95 ± 5.75 *	64.26 ± 22.68 *
	*p* values	0.396		**0.037**	
2	Necrosis	13.04 ± 5.76	8.56 ± 13.40	13.04 ± 5.76 *	26.76 ± 6.32 *
	*p* values	0.470		**0.030**	
3	Incipient apoptosis	0.10 ± 0.01 *	7.08 ± 10.46 *	0.10 ± 0.01	6.67 ± 10.14
	*p* values	**0.025**		0.105	
4	Late apoptosis	0.00 ± 0.00	2.96 ± 6.55	0.00 ± 0.00	2.28 ± 5.58
	*p* values	0.115		0.286	

Legend: X, obtained results mean; SD, standard deviation; PCa, prostate adenocarcinoma; BPH, benign prostatic hyperplasia; IA, incipient apoptosis; LA, late apoptosis; * *p* < 0.05 represents a statistically significant difference between controls and experimental samples made by independent *t* test.

**Table 5 biomedicines-10-01672-t005:** Evaluation of T lymphocyte biomarkers and intratumoral expression of CD34 in prostate adenocarcinoma, benign prostatic hyperplasia, and non-malignant adjacent tissue samples.

Number	Parameters	Control X ± SD	PCa X ± SD	Control X ± SD	BPH X ± SD
1	CD3+	44.00 ± 22.62	59.00 ± 22.43	44.00 ± 22.62	57.70 ± 25.48
	*p* values	0.386		0.499	
2	CD4+	66.00 ± 48.08	54.68 ± 21.09	66.00 ± 48.08	66.00 ± 26.11
	*p* values	0.533		0.919	
3	CD8+	45.00 ± 28.28	31.25 ± 17.55	45.00 ± 28.28	26.00 ± 14.49
	*p* values	0.334		0.166	
4	CD34+	1.50 ± 0.70 **	26.12 ± 6.84 **	1.50 ± 0.70	2.60 ± 1.07
	*p* values	**0.000**		0.204	

Legend: X, obtained results mean; SD, standard deviation; PCa, prostate adenocarcinoma; BPH, benign prostatic hyperplasia; CD3, total lymphocytes; CD4, T helper lymphocytes; CD8, cytotoxic T lymphocytes; CD34, intratumoral CD34 glycoproteins; ** *p* ≤ 0.01 represents statistically significant differences between controls and experimental samples made by independent *t* test.

**Table 6 biomedicines-10-01672-t006:** DNA content and adhesion glycoproteins expressions correlations.

Cell Cycle	CD34 + CD61+	CD61+	CD61-	CD42b+	CD42b-
G0/G1 phase	−0.514 **	−0.150	0.121	−0.475 *	0.490 **
*p* values	**0.005**	0.447	0.539	**0.011**	**0.008**
S phase	−0.236	−0.430 *	0.431 *	−0.121	0.151
*p* values	0.226	**0.023**	**0.022**	0.539	0.444
G2/M phase	0.513 **	0.226	−0.200	0.446 *	−0.470 *
*p* values	**0.005**	0.247	0.307	**0.017**	**0.012**

Legend: * *p* < 0.05 and ** *p* < 0.01 represent statistically significant differences between cases made by Pearson correlations.

**Table 7 biomedicines-10-01672-t007:** Correlations between glycoprotein levels implied in cell proliferation.

Glycoproteins	CD61T	CD61+	CD61-	CD34T	CD34+	CD34-	CD42b+	CD42b-
**CD34+ CD61+**	0.512 *	0.589 *	−0.550 *	0.503 *	0.500 *	−0.516 *	0.623 *	−0.641 *
*p* values	**0.005**	**0.001**	**0.002**	**0.006**	**0.007**	**0.005**	**0.000**	**0.000**

Legend: * *p* < 0.01 represents a statistically significant difference between cases made by Pearson correlations.

**Table 8 biomedicines-10-01672-t008:** Tissue inflammation grade (TIG) correlated with adhesion glycoproteins expressions, cell viability, and late cell apoptosis.

Parameters	CD61+	CD61-	CD42b+	CD42b-	Viability	LA
**TIG**	0.544 *	−0.535 *	0.664 **	−0.639 **	−0.632 *	0.528 *
*p* values	0.024	0.027	0.004	0.006	0.011	0.043

Legend: ** *p* < 0.01 and * *p* < 0.05 represent statistically significant differences between cases made by Pearson correlations; LA, late apoptosis.

**Table 9 biomedicines-10-01672-t009:** Ki-67 cell proliferation correlates with viability, necrosis, and intratumoral CD34 expression.

Parameters	Viability	Necrosis	CD34 IT
**Ki-67 expression**	0.548 *	−0.682 **	0.611 **
*p* values	0.034	0.005	0.009

Legend: ** *p* < 0.01 and * *p* < 0.05 represent statistically significant differences between cases made by Pearson correlations; CD34 IT, intratumoral CD34 glycoproteins expression.

## Data Availability

Data are contained within the article.

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
