# Peer review of "Characterization of the Tumor Microenvironment and the Biological Processes with a Role in Prostatic Tumorigenesis"

_biomedicines, 2022, doi:10.3390/biomedicines10071672_

Round 1

Reviewer 1 Report

This is an interesting study that  characterize the tumor microenvironment (T lymphocytes 26 infiltration, intratumoral CD34, and KI-67 expressions) by immunohistochemistry methods and 27 study the biological mechanisms (cell cycle, cell proliferation by adhesion glycoproteins, cell apop- 28 tosis) involved in the evolution of the prostate tumor process , I suggest to increase the clinical appeal of your study defining:

-        Potential role in selction of patients elected for Active surveillance , define interaction with liquid biopsy and other biomarkers  , see : Minerva Urol Nephrol. 2021 Aug;73(4):442-451. doi: 10.23736/S2724-6051.21.04098-4. Epub 2021 Mar 26.; ; ;

-        Prostate Cancer and Prostatic DiseasesVolume 22, Issue 1, Pages 101 - 1091 March 2019

-        Ability to predict bcr and metastasis after curative treatment for PCa , as previously showing for other genomic test

-        Define the strong correlation with Isup or GS and therapeutic response to cht as Future OncologyVolume 9, Issue 6, Pages 889 - 897June 2013

-         

Author Response

Dear Reviewer, 1, 

Thank you for your appreciation and recommendations. As you suggested, we modified the manuscript and included the important references to increase the clinical appeal.

“Our selection criteria were applied to identify and establish the clinical efficiency of the human PCa and BPH biomarkers by flow cytometry and IHC methods, highlighting the characterization of the tumor microenvironment, in conformity with references [25].”

“Biochemical recurrence of prostate cancer (bcr) and metastasis prediction after PCa curative treatment were showed by references for other molecular tests [65].“

“The authors observed a strong correlation between GS and therapeutic response to cabazitaxel in metastatic castration-resistant prostate cancer patients [81].”

The bibliography was modified and introduced the sources of the cited references.

I revised the manuscript and improved the English language and style, as your suggestion.

Yours faithfully,

Ph.D. Biologist Matei Elena

 [email protected]

Reviewer 2 Report

The study of micro environment is important for prostate cancer. There are some critical issues regarding the methods:

1: since prostate cancer is heterogenous, macro dissection is difficult to purify tumor and normal parts , some cancer part still have BPH TISSUE, ideally, micro dissection with laser captures micro dissection is needed. Authors should discuss this issue

2: the cell line co-culture system should be consider for more detail mechanism study

Author Response

Dear Reviewer, 2,

Thank you for your appreciation and recommendations. As you suggested, because prostate cancer is heterogeneous, macro dissection is difficult to purify tumor and non-malignant parts, because some cancer part still has BPH tissue. For ideal cases, micro dissection with laser captures may be needed. We agreed with your recommendation, but in Romania, we did not use the micro dissection with the laser to separate the tumoral from the non-malignant parts of tissue samples. In the Clinical Service of Pathology from our hospital, the macro dissection of tumoral and non-malignant parts of the tissue is made by experienced pathologists. After their examination, we received the tissue samples in the laboratory of Cell Biology from the research center of our university and worked the flow cytometry parameters (cell cycle, adhesion molecules implied in cell proliferation, and cell apoptosis). We introduced in the manuscript the micro dissection with laser capture as a limitation of our study.

As you suggested, the cell line co-culture system should be considered to detail the mechanisms represents an important recommendation for us and will be used as the future direction of our research to observe the biological processes implied in the prostate tumor microenvironment characterization. This recommendation will be mentioned in the manuscript, such as future directions to develop it.

Design and objectives are developed in our grant, and we received the funds to buy the necessary materials to support our research theme. This manuscript may represent a result indicator and will be reported in our grant (PROMETEU grant, contract number 06/October 20th, 2021).

We modified the conclusions in the manuscript, remaining only a general conclusion about the importance of these biomarkers in the study of the PCa and BPH tissue samples. „Biological mechanisms implied in the prostate tumor microenvironment characterization represented by the cell cycle, apoptosis, adhesion glycoproteins, T lymphocytes infiltrations, Ki-67, and intratumoral CD 34 biomarkers provide efficient means of measurement by flow cytometry and IHC techniques for PCa and BPH tissue samples and should be explored in the future not only for the diagnostics but also for the therapeutic purposes.”

 As your suggestion, I revised the manuscript and improved the English language and style.

Yours faithfully,

Ph.D. Biologist Matei Elena

 [email protected]

Reviewer 3 Report

In this study, samples of treatment naive cases of benign prostatic hyperplasia (BPH) and prostates cancer (PCa) have been evaluated using IHC and flow cytometry for a few limited markers. Based of cell cycle profiles and the studies biomarkers a series of correlative results are listed. In my review, these are valuable observation while it is not enough for an original article. My major concern is the research design and the objectives of this work. The conclusions are not supported by the results, and they are speculative. Therefore, I cannot recommend this work to be considered for publication.  

Author Response

Dear Reviewer, 3,

Thank you for your appreciation and recommendations. We have studied in this manuscript the important biological mechanisms implied in the tumor cell proliferation, and we agree with you, that these are valuable observations made in our laboratory. Design and objectives are developed in our grant, and we received the funds to buy the materials which are necessary to support our research theme. All projects include the articles as performance indicators in the research. This manuscript may represent a result indicator and will be reported in our grant (PROMETEU grant, contract number 06/October 20th, 2021).

Our obtained results are important and unpublished observations representing our work as researchers and giving us the right to publish an original article. In cancer area research, you know that any information is valuable (articles, projects), and studying the biological mechanisms implied in the tumoral microenvironment characterization, represents the first step to knowing more about these because all prostate tissue samples are recovered from patients without applied treatments, and the studied biomarkers represent more than valuable observations and may provide the future research directions not only for the diagnostics but also for the therapeutic purposes.

Also, new references were introduced from experienced researchers in the prostate cancer field.

As you suggested, we modified the manuscript’s conclusions to eliminate the speculative interpretation. A general conclusion about the importance of these biomarkers in the study of the PCa and BPH tissue samples remains. „Biological mechanisms implied in the prostate tumor microenvironment characterization represented by the cell cycle, apoptosis, adhesion glycoproteins, T lymphocytes infiltrations, Ki-67, and intratumoral CD 34 biomarkers provide efficient means of measurement by flow cytometry and IHC techniques for PCa and BPH tissue samples and should be explored in the future not only for the diagnostics but also for the therapeutic purposes.”

Yours faithfully,

Ph.D. Biologist Matei Elena

 [email protected]

Round 2

Reviewer 3 Report

This revision includes a minor revision on a few paragraphs while the data, figures, structure and rationales are same as before. Unfortunately, I cannot recommend this work for further review. 

Author Response

Dear Reviewer 3,

Thank you for the comments. About your propositions, we told you that our manuscript design respects the objectives of our project, which was won in the grant competition and represents a result indicator of this project.

About the data and figures used in the manuscript, how do you want to improve them? Data are original, obtained from patients with prostate cancer and patients with benign hyperplasia, and we cannot change the data because it represents the diagnostics of the patients. To these patients were not applied treatments and for us was an opportunity to study the usual biomarkers for IHC such as T lymphocytes infiltration, intratumoral CD34, and ki-67 expressions, and usually flow cytometry biomarkers such as cell cycle, cell apoptosis, and few adhesion glycoproteins as CD61 which are specifically for T cells and CD42b which are specifically for platelets.

Our reasoning was straightforward because we observed correlations between cell biology processes and pathology, which are implied in the characterization of the tumoral microenvironment. These biomarkers are valuable to be published because they can give new research directions to improve the diagnostics and therapeutics, being the rapid and efficient methods used in the laboratory.

Also, we cannot change the figures used in the manuscript because they are original, obtained from the soft of the calculator of the flow cytometer, being the graphic representations of the applied biomarkers on our samples.

Improvement of our data, figures, which are originals is not well, because what you recommend us, maybe also interpretable. 

In a previous revision, the conclusions were modified to be representative of our manuscript, and the references were added to the manuscript.

Yours faithfully,

Ph.D. Biologist Matei Elena

 [email protected]
